# Gait Training with Virtual Reality-Based Real-Time Feedback for Chronic Post-Stroke Patients: A Pilot Study

**DOI:** 10.3390/healthcare13020203

**Published:** 2025-01-20

**Authors:** Sunmin Kim, Yangjin Lee, Kyunghun Kim

**Affiliations:** 1Department of Physical Therapy, Gimcheon University, Gimcheon 39528, Republic of Korea; jjssppaarrkk@hanmail.net; 2Department of Physical Therapy, Kyungbuk College, Yeongju 36133, Republic of Korea; ptyangjin2@naver.com; 3Department of Physical Therapy, Daejeon Health University, Daejeon 30711, Republic of Korea

**Keywords:** virtual reality, real-time feedback, balance, gait

## Abstract

Background: Virtual reality-based training has been widely used for post-stroke patients due to its positive effects on functional aspects by promoting brain plasticity. Objective: This study aimed to investigate the effectiveness of gait training with virtual reality-based real-time feedback on motor function, balance, and spatiotemporal gait parameters in post-stroke patients. Methods: Fifteen patients (*n* = 15) with chronic stroke were randomly assigned to either the virtual reality-based real-time feedback with treadmill gait training (experimental group *n* = 8) or the treadmill gait training (control group *n* = 7). For the experimental group that participated, a treadmill, an Oculus Rift VR device, and smart insoles were used for gait training with VR-based real-time feedback. Regarding gait training with VR-based real-time feedback, the patient wore an Oculus Rift and performed gait training on a treadmill for 30 min a day, three times a week, for 5 weeks. The control group participated in treadmill gait training for 30 min a day, three times a week, for 5 weeks. Motor function was measured using the Fugl-Meyer assessment. Balance was measured using the timed up and go test and Berg balance scale. Gait performance was measured using an Optogait. The normality test was performed using the Shapiro–Wilk test, the Wilcoxon signed-rank test was used for the within-group comparison, and the Mann–Whitney U test was used for the between-group comparison. Results: In the group analyses, both groups showed significant improvements in motor function balance and gait ability. According to the pre- and post-treatment results, greater improvement in the Fugl-Meyer assessment (experimental group: 4.75 vs. control group: 1.57) was observed in the experimental group compared with the control group (*p* < 0.05). In balance ability, greater improvement in the timed up and go test (experimental group: −3.10 vs. control group: −1.12) and Berg balance scale (experimental group: 3.00 vs. control group: 1.71) (*p* < 0.05). In the spatiotemporal gait parameters, greater improvement in affected step length (5.35 vs. 2.01), stride length (3.86 vs. 1.75), affected single support (2.61 vs. 1.22), and cadence (0.07 vs. 0.02) was observed in the experimental group compared with the control group (*p* < 0.05). Conclusions: This study suggested the positive effects of the virtual reality-based real-time feedback with treadmill gait training on motor function, balance, and gait performance.

## 1. Introduction

Individuals who have experienced a stroke encounter various disabilities depending on the specific patterns of their condition, which can significantly impact their daily lives [1]. Disabilities resulting from a stroke can include motor impairments, sensory deficits, and cognitive dysfunction [2]. These disabilities can lead to weakened postural control abilities, resulting in balance and gait disturbances [3].

Numerous studies have been conducted to address the functional performance issues faced by post-stroke patients. Effective physical therapy methods for enhancing functional recovery include Proprioceptive Neuromuscular Facilitation (PNF), Bobath therapy, Rood therapy, Constraint-Induced Movement Therapy (CIMT), task-oriented training, and virtual reality [4,5,6,7]. These therapies enable individuals to freely adjust activities from abnormal movement patterns, and their effectiveness in enhancing functional performance has been demonstrated in numerous studies [8].

Balance is an essential element for independence [9], and balance deficits resulting from stroke can arise from various factors, including sensory deficits and vestibular dysfunction [10]. Following a stroke, individuals may struggle to maintain their body’s center and experience issues with task-oriented movements, leading to an increased risk of falls [11].

Gait is one of the keys to functional independence, serving as an important element for moving from one location to another within the environment [12]. Among patients who experience gait disturbances due to stroke, 22% fail to regain walking function [13]. The gait characteristics associated with stroke can include reduced walking speed [14], circumnutated gait [15], and excessive knee extension [16]. Treadmill-based intervention methods have been used in various ways in post-stroke patient rehabilitation to improve walking disorders, but simple walking movements have the disadvantage of being boring and lacking motivation for patients [17].

Recent research has shown an increased interest in the use of technologies like virtual reality for stroke rehabilitation. Virtual reality can be employed to improve patients’ upper limb function, balance, gait, overall motor function, and cognitive function [18]. Compared to traditional rehabilitation therapies, the most significant advantage of virtual reality rehabilitation is that post-stroke patients perceive the treatment as engaging and game-like, which can enhance their attention, motivation, and adherence to therapy [19]. Additionally, virtual reality therapy combined with telemedicine has also been widely used recently, and this approach allows patients to receive treatment at home while medical professionals can directly monitor the patient’s condition and provide feedback in real time [20]. Specifically, virtual reality training for post-stroke patients has been reported as a safe and cost-effective rehabilitation method for improving lower limb function, balance, stair climbing, ankle strength, range of motion, and walking speed [21,22].

Virtual reality training utilizes specialized equipment to recreate similar experiences to those the user currently encounters, offering a simulation technique [23]. Virtual reality can be broadly categorized into non-immersive and fully immersive systems based on the level of immersion [24]. Non-immersive virtual reality is the most common experience in daily life and refers to systems that interact with devices like smartphones, computers, and televisions, using controllers such as a mouse or keyboard [25]. Fully immersive virtual reality employs head-mounted displays to eliminate real sensory information and create realistic experiences based on computer-generated information [26].

Training using virtual reality is effective in enhancing neuroplasticity and relearning motor patterns and skills [27]. Research has demonstrated improvements in measures such as the 10-m walk test, timed up and go test [28], stride time, and cadence [29]. Additionally, ongoing reports detail results from studies combining various interventions, including treadmill [4], robotic devices [30], and EMG [31]. However, most of the previous studies are on virtual reality, and there is still a lack of research results on balance and gait training using real-time feedback and virtual reality, and in particular, research on motor function, balance, and gait is lacking [31].

Therefore, this study aims to investigate the effects of a real-time feedback-based gait virtual reality program on the spatial–temporal parameters of motor function, balance, and gait in chronic post-stroke patients.

## 2. Materials and Methods

### 2.1. Research Design

This study utilized a single-blind, random pre-test and post-test control group pilot experimental design. A physical therapist with over five years of clinical experience and a master’s degree conducted the measurements and experiments without being aware of the group assignments for each patient. The therapist instructed the subjects to stop the experiment immediately and get enough rest if they felt dizzy or had other symptoms that may occur during the experiment.

This study is registered with the Clinical Research Information Service database and has received approval from the university’s Institutional Review Board (GU-202207-HRa-05-01-P). This study was conducted over a period of five weeks, from 1 May 2023 to 2 June 2023, following IRB approval. All the participants read the informed consent document and voluntarily agreed to participate in this study, providing written consent prior to registration. Although this is a sponsored study, the researchers designed and conducted the research, including the data analysis, manuscript drafting, revision, and submission, without any intervention from the funding source.

### 2.2. Subjects

This study involved 15 stroke patients who received outpatient treatment at Hospital B in Seongnam, Gyeonggi Province.

This study was conducted with participants who agreed to participate after being introduced to and informed about the research through a recruitment announcement. The participants were outpatients who had experienced a stroke for more than six months.

The participants for this study were initially recruited by posting a study participation poster in the physical therapy room. Those who expressed interest in participating were provided with an explanation of this study and were asked to sign an informed consent form before being selected as study participants. The recruitment criteria for VR-based interventions focused on patients with gait problems that the VR system aimed to improve, such as reduced stride length and balance issues. Since this study has limitations in applying VR-based interventions to patients with severe gait impairments, individuals with a Modified Ashworth Scale (MAS) score of 2 or lower and those capable of minimal independent walking were selected. Initially, 18 stroke patients expressed their willingness to participate in this study. However, after evaluating their suitability for VR interventions, a final group of 15 participants was selected. The inclusion and exclusion criteria for this study are as follows.

The specific inclusion criteria for the research participants are as follows:

First, the participants must have experienced a duration of illness lasting at least six months and be capable of independent ambulance at a minimum level. Second, they must have a Functional Ambulation Classification (FAC) score of 3 or less. Third, they should score 21 or higher on the Korean mini-mental state examination, indicating adequate communication abilities. Fourth, the participants must have no allergic reactions to functional electrical stimulation. Lastly, they should not have any walking issues due to ankle joint contractures.

The exclusion criteria for the research participants are as follows:

Individuals with symptoms of vestibular dysfunction or cerebellar disorders, those with visual or auditory impairments, individuals who have undergone orthopedic surgery on the lower extremities, and those with cardiovascular issues or other neurological symptoms affecting the lower limbs were excluded from this study.

### 2.3. Procedure

All the research participants were assessed for motor function, balance, and gait ability both before the intervention and after 5 weeks of treatment. The selected participants received an explanation of the experimental procedures before this study commenced. The participants were randomly assigned to either the experimental group or the control group in a 1:1 ratio using a computer-generated random number table. All the participants engaged in the intervention for 30 min a day, 5 days a week, totaling 15 sessions. Both groups also participated in a standard rehabilitation program, consisting of 30 min per session, once a day, 5 days a week.

### 2.4. Research Methods

#### 2.4.1. Experimental Group

In this study, the experimental group performed gait training on a treadmill while wearing an Oculus Rift and smart insoles (SPINA Systems Co., Ltd., R-C-SPO-Pedisol250, Republic of Korea). The experimental group conducted real-time feedback virtual reality gait training on the treadmill (HP Cosmos, QUASAR MED, Germany) while wearing the Oculus Rift. During the gait training, they visually confirmed real-time feedback information obtained from the smart insoles. The real-time feedback was provided after confirming that the sensors were working correctly by pressing the calibration key while standing and then starting the gait training.

The participants performed the gait training while receiving real-time feedback on the displacement of the center of pressure (COP) and the weight-bearing status of both the unaffected and affected sides. This feedback was visually represented with bar graphs on both sides, indicating the degree of weight shift and load, helping the participants properly shift and bear weight on both the unaffected and affected sides.

The participants were given sufficient explanations and a 10 min practice session before the experiment to ensure they could adequately adapt to the real-time feedback virtual reality training. The walking speed was set to a pace that the participants found comfortable. The virtual reality gait training involved pre-recording an outdoor walking environment in 3D and uploading it in an online format, which the participants then used for gait training through the VR device.

The virtual reality program sessions were set to include 5 min of walking in a park, 5 min of walking in the city, and 5 min of shopping. After each 5 min virtual reality training session, the participants took a 5 min rest to prevent muscle fatigue and dizziness before proceeding to the next session [32]. The real-time feedback virtual reality training was applied for a total of 15 min, while conventional physical therapy was conducted for 30 min.

This study was conducted by a physical therapist with over 10 years of experience and certification in neurodevelopmental treatment, who directly administered the VR-based intervention.

The real-time feedback virtual reality training was conducted for 30 min a day, three times a week, for a total of 15 sessions over 5 weeks. Conventional physical therapy was conducted for 30 min a day, five times a week, for 5 weeks. Both the real-time feedback virtual reality training and the traditional physical therapy were administered by the same physical therapist, who had three years of clinical experience and a master’s degree. The traditional physical therapy included Bobath, PNF (Proprioceptive Neuromuscular Facilitation), range-of-motion exercises, limb strength, stretching exercises, ground walking training, and cycling exercises.

#### 2.4.2. Control Group

The control group performed the gait training on a treadmill under the same conditions as the experimental group but without the use of virtual reality. The treadmill training consisted of 5 min of training followed by 5 min of rest, totaling 3 sets. Additionally, conventional physical therapy was conducted for 30 min alongside the 30 min of treadmill gait training.

### 2.5. Measurement Items

#### 2.5.1. Motor Function

The Fugl-Meyer assessment (FMA) was conducted to evaluate the motor function of the lower limbs. It consists of 17 items, with a scoring range of 0 to 34 points. The assessment items include reflex activity; coordination abilities of the hip, knee, and ankle joints; and voluntary movement. The inter-rater reliability for post-stroke patients was high, with an r ≥ 0.96, and the concurrent validity was also high, with an r ≥ 0.99, making it a reliable assessment tool [33].

#### 2.5.2. Balance Ability

The Berg balance scale was conducted to assess the balance ability of the post-stroke patients. The assessment items are categorized into three areas, sitting, standing, and changing positions, consisting of a total of 14 items. Each item is scored from 0 to 4 points, with a maximum score of 56 points; a higher score indicates better balance ability. The intra-rater reliability was r = 0.99, and the inter-rater reliability was r = 0.97 [34,35]. The timed up and go test (TUG) was used to evaluate the balance ability of the post-stroke patients. Using a chair with armrests, the test begins with the participant seated, and they are instructed to stand up on command, walk 3 m to a turning point, and return to sit in the chair. Prior to the test, the participants were educated to turn toward the unaffected side at the turning point. Measurements were taken three times, and the average value was used for analysis. The intra-rater reliability was r = 0.99, and the inter-rater reliability was r = 0.98 [36].

#### 2.5.3. Gait Ability

The gait assessment was conducted using the OptoGait analyzer (Microgate S.r.l, Italy) to evaluate the spatiotemporal characteristics of walking in post-stroke patients. The gait analyzer consists of two transmitting and receiving bars, each 4 m long, and a webcam Pro 9000 (Logitech, Switzerland). Both bars are installed on the ground with a width of 1 m. Inside each bar, light-emitting diodes (LEDs) are placed at 1 cm intervals, and communication occurs via infrared signals continuously sent from the transmitting bar to the receiving bar. The participants were instructed to walk 10 m at a comfortable pace on a flat surface, with the first 2 m and the last 2 m excluded from the measurement. During the 6 m walking segment, the participant’s footfalls were detected, and information about gait variables was collected. The spatial–temporal characteristics of the gait analyzed in this study included the following: weight-bearing on the affected side, single support time on the affected side, double support time, step length, and velocity. All the measurements were conducted by a single skilled physical therapist to eliminate inter-rater variability. The intra-rater reliability was found to be r = 0.99, and the test–retest reliability ranged from r = 0.98 to 0.99 [37].

### 2.6. Data Analysis

The number of subjects in this study was determined based on the results of previous studies. Using the G-Power program (G-power version 3.1.9.4), the sample size was calculated with a power of 0.8, a significance level of 0.05, and an effect size of 0.723, resulting in a total of 15 participants.

All the statistical analyses in this study were conducted using SPSS 21.0. The Shapiro–Wilk test was employed to assess normality. Independent sample *t*-tests and Chi-square tests (for categorical variables) were performed to compare the general characteristics of the participants and the homogeneity between the two groups prior to the intervention. The Wilcoxon signed-rank test was conducted to compare the pre- and post-rehabilitation training-dependent variables within each group. Additionally, the Mann–Whitney U test was utilized to compare the differences in the change in dependent variables between the training methods across the groups. The significance level (α) for all the statistical tests was set at 0.05 or lower.

## 3. Results

### 3.1. General Characteristics of the Subjects

The general characteristics of the subjects are as follows (Table 1). The experimental group had an average age of 52.75 ± 10.70 years, while the control group had an average of 51.71 ± 7.67 years, with no statistically significant difference between the two groups (*p* > 0.05). The average height of the subjects in the experimental group was 166.40 ± 3.76 cm, and in the control group, it was 166.94 ± 7.94 cm, with no statistically significant difference between the groups (*p* > 0.05). The average weight of the subjects in the experimental group was 67.63 ± 6.41 kg, and in the control group, it was 72.18 ± 9.26 kg, with no statistically significant difference between the groups (*p* > 0.05). The gender of the subjects in the experimental group was male 3 and female 5 subjects, and in the control group, it was male 4 and female 3 subjects, with no statistically significant difference between the groups (*p* > 0.05).

### 3.2. Change in Motor Function According to Intervention

Significant differences were observed in the motor function abilities before and after training in both groups. The difference in the Fugl-Meyer assessment scale within the two groups was statistically significant, favoring the experimental group (*p* < 0.05). Comparing the change in the Fugl-Meyer assessment scale between the two groups also showed a statistically significant difference, favoring the experimental group (*p* < 0.05) (Table 2).

### 3.3. Change in Balance According to Intervention

Significant differences were observed in balance abilities before and after training in both groups. When comparing the change in the BBS and TUG within the two groups, the experimental group showed a statistically significant difference compared to the control group (*p* < 0.05). Comparing the change in the BBS and TUG between the two groups revealed a statistically significant difference, favoring the experimental group (*p* < 0.05) (Table 3).

### 3.4. Change in Spatiotemporal Gait Parameters According to Intervention

Significant differences were observed in the spatiotemporal gait parameters before and after training in both groups. When comparing the change in the ASL, SL, ASS, and cadence within the two groups, the experimental group showed a statistically significant difference compared to the control group (*p* < 0.05). Comparing the change in the ASL, SL, ASS, and cadence between the two groups revealed a statistically significant difference, favoring the experimental group (*p* < 0.05) (Table 4).

## 4. Discussion

Generally, the primary goal for post-stroke patients is to perform walking and daily activities independently. The purpose of this study was to investigate the effects of a real-time feedback-based walking virtual reality program on motor function, balance, and spatiotemporal parameters of gait in patients with chronic stroke. This study was conducted as a single-group trial with 15 chronic post-stroke patients, implementing the program for 30 min a day, three times a week, over a total of five weeks. The aim was to evaluate the utility of real-time feedback virtual reality treadmill walking training. The main findings indicated that the experimental group showed significant improvements in motor function, balance, and gait compared to the control group.

In this study, while both the experimental and control groups exhibited improvements in all aspects, the experimental group showed significant differences in motor function compared to the control group. According to a study by Anwar et al. (2021) [38], involving 68 post-stroke patients who underwent virtual reality and conventional physical therapy three times a week for six weeks, the virtual reality group demonstrated significant improvements in the Fugl-Meyer score for motor function, joint pain, and range of motion compared to the conventional therapy group. Similarly, a study by Yaman et al. (2022) [39] involving 60 patients with chronic stroke found that the experimental group, which received 30 min of virtual reality training followed by 30 min of conventional physical therapy, showed greater improvements than the control group, which underwent 60 min of conventional therapy, in the Fugl-Meyer assessment—lower extremity and balance ability. These results align with the findings of the current study.

These results suggest that real-time feedback virtual reality training enhances weight-bearing ability and activates the muscles around the hip joint, thereby improving motor function of the lower limbs. Furthermore, combining treadmill walking training allows for simultaneous sensory input, enabling a focus on the signals needed for walking and strength in the lower limbs, which likely contributed to the improvement in lower limb function.

Therefore, this study indicates that a real-time feedback-based walking virtual reality program is effective in improving motor function, balance, and gait abilities in patients with chronic stroke. These findings may serve as important foundational data for the development of rehabilitation programs for post-stroke patients in the future.

In this study, while both the experimental and control groups showed improvements in all areas, the experimental group exhibited a significant difference in balance compared to the control group. In a study by Karasu et al., balance training based on Nintendo Wii was conducted on 23 selected participants from a total of 70 post-stroke patients. The control group received conventional physical therapy, occupational therapy, and neurodevelopmental therapy, with both groups participating in interventions for a total of 8 weeks, 5 times a week, for 2 to 3 h per day. The balance training based on Nintendo Wii measured weight and center of pressure through the Wii Balance Board, providing feedback on their movements, and involved six out of nine balance training exercises. As a result, the experimental group showed statistically significant improvements in the BBS, FRT, TUG, PASS, and Static Balance Index.

In another study by Porras et al., a total of 167 participants were included, categorized into six groups: Parkinson’s disease (36), stroke (31), multiple sclerosis (9), traumatic brain injury (10), other diseases (42), and non-neurological diseases (39). Each group underwent a total of 12 sessions consisting of interventions related to balance, walking, and cognition, each lasting 30 to 45 min. Among these, there were sessions that utilized virtual reality in more than two instances. The interventions were designed to suit the individual characteristics of the participants and included task-oriented training, sensory feedback, and motor relearning through virtual reality training. As a result, statistically significant differences were observed in the 10 m walking test, TUG, and BBS.

In a study by Silva et al. (2015) [40], a comparison was made between the walking abilities of post-stroke patients in real life and in virtual reality by conducting a 3 min stationary walking test with 10 post-stroke patients. The results indicated that walking ability, particularly in terms of cadence and ankle dorsiflexion angle, improved more in virtual reality. In the research by Cho and Lee (2019) [41], the effects of non-immersive virtual reality-based treadmill training were compared with those of conventional treadmill training among 30 chronic post-stroke patients. The intervention lasted a total of 6 weeks, with sessions held 3 times a week for 30 min each day. The non-immersive virtual reality-based treadmill training involved viewing videos of walking in various real environments while conducting treadmill training.

As a result, statistically significant improvements were observed in the Berg balance scale (BBS) and timed up and go (TUG) tests during the dynamic balance assessment, as well as in the temporal gait parameters, including gait speed, cadence, single-limb support period, and double-limb support period. Spatial gait parameters such as step length and stride length also showed statistically significant improvements. Notably, when compared to conventional treadmill training, significant differences between groups were found in the BBS, TUG, gait speed, cadence, single-limb support period, step length, and stride length.

Additionally, in a study by Enam et al., treadmill training using augmented reality was conducted with two post-stroke patients. The intervention period lasted a total of four weeks, with sessions conducted three times a week, and evaluations were carried out before, after, and one month post-intervention. As a result, significant improvements were noted in the BBS, 10 m walk test, and 6 min walk test. This suggests that the treadmill gait training led to increased lower limb strength, and the real-time feedback facilitated easier weight shifting and support on the affected side, thereby improving balance and gait speed.

Furthermore, the real-time feedback-based virtual reality treadmill walking training, through consistent effort, practice, and repetition, enhanced motor learning abilities. It also improved balance during the stance phase on the affected side, which led to better weight shifting and an enhancement in the spatiotemporal aspects of walking.

The research results obtained from this study have several limitations. First, the subjects were limited to chronic post-stroke patients who met the selection criteria, making it difficult to generalize the findings to all post-stroke patients. Second, the time it took for each patient to adapt to wearing the virtual reality device varied according to their condition, and there were instances where the intervention was interrupted due to dizziness. Third, both the experimental group and the control group received general physical therapy in addition to the included interventions, which may have influenced the research results. Fourth, the sample size was small, consisting of only 15 participants, and the severity of the patients’ strokes and the degree of spasticity were not carefully assessed in the analysis. Future studies should include a larger sample size and an assessment of spasticity to better generalize the effectiveness of virtual reality interventions for stroke patients.

In the future, we believe that by addressing these limitations, training programs using real-time feedback-based virtual reality can be applied not only to maintain balance but also to various task-oriented training and dual-task training during walking. This approach can be applied to patients with diverse lesions, not limited to chronic post-stroke patients. However, there are currently no virtual reality-related medical devices or rehabilitation training programs for patients, highlighting the need for ongoing research and development in this area.

## 5. Conclusions

This study investigated the effects of treadmill walking training combined with real-time feedback virtual reality on lower limb motor function, balance, and walking ability. The results confirmed that treadmill walking training with real-time feedback virtual reality was effective in improving motor function, balance, and walking ability compared to the treadmill walking training group. Therefore, it is believed that this approach can be applied as a component of training programs aimed at enhancing motor function, balance, and walking ability in post-stroke patients.

## Figures and Tables

**Table 1 healthcare-13-00203-t001:** General characteristics of subjects (N = 15).

	Experimental Group (*n* = 8)	Control Group (*n* = 7)	*t*/x^2^ (*p*)
Age (year)	52.75± 10.70 ^a^	51.71 ± 7.67	−0.212 ^b^ (0.835 ^c^)
Weight (kg)	67.63 ± 6.41	72.18 ± 9.26	−1.118 (0.284)
Height (cm)	166.40 ± 3.76	166.94 ± 7.94	−0.173 (0.865)
MMSE-K (score)	26.75 ± 0.71	27.00 ± 0.82	−0.636 (0.536)
Gender (male/female)	3/5	4/3	0.579 (0.447)
Onset (month)	7.50 ± 7.55	7.71 ± 7.55	−0.548 (0.593)
Affected side (right/left) Stroke type (Ischemic/Hemorrhagic)	4/4	5/2	0.714 (0.398)

Note. ^a^ Mean ± SD, ^b^ Chi-square test, and ^c^ independent *t*-test. MMSE, mini-mental state examination.

**Table 2 healthcare-13-00203-t002:** Change in motor function according to intervention.

		Experimental Group (*n* = 8)	Control Group (*n* = 7)	z	*p*
FMA	Pre	21.62 ± 2.13 ^a^	20.57 ± 1.98		
	Post	26.37 ± 0.74	22.14 ± 1.95		
	Post–Pre	4.75 ± 1.58	1.57 ± 0.53	−3.105	0.002 *
	z (*p*)	−2.552 (0.011 *)	−2.428 (0.015 *)		

Note. ^a^ Mean ± SD, FMA: Fugl-Meyer assessment, and * *p* < 0.05.

**Table 3 healthcare-13-00203-t003:** Change in balance according to intervention.

		Experimental Group (*n* = 8)	Control Group (*n* = 7)	z	*p*
BBS	Pre	44.50 ± 2.72 ^a^	44.14 ± 2.41		
	Post	47.50 ± 2.56	45.85 ± 1.86		
	Post–Pre	3.00 ± 0.75	1.71 ± 0.95	−2.304	0.021 *
	z (*p*)	−2.558 (0.011 *)	−2.414 (0.016 *)		
TUG	Pre	16.19 ± 1.68	17.19 ± 1.73		
	Post	13.09 ± 1.15	16.07 ± 1.77		
	Post–Pre	−3.10 ± 1.60	−1.12 ± 0.43	−3.125	0.002 *
	z (*p*)	−2.521 (0.012 *)	−2.366 (0.018 *)		

Note. ^a^ Mean ± SD, BBS: Berg balance scale, TUG: timed up and go test, and * *p* < 0.05.

**Table 4 healthcare-13-00203-t004:** Change in spatiotemporal gait parameters according to intervention.

		Experimental Group (*n* = 8)	Control Group (*n* = 7)	z	*p*
ASL	Pre	41.87 ± 6.85 ^a^	43.97 ± 7.47		
	Post	47.22 ± 6.95	45.98 ± 7.14		
	Post–Pre	5.35 ± 3.26	2.01 ± 1.23	−2.609	0.009 *
	z (*p*)	−2.524 (0.012 *)	−2.366 (0.018 *)		
SL	Pre	64.22 ± 7.71	68.47 ± 8.93		
	Post	68.08 ± 7.41	70.22 ± 7.89		
	Post–Pre	3.86 ± 2.07	1.75 ± 1.34	−2.269	0.023 *
	z (*p*)	−2.524 (0.012 *)	−2.384 (0.017 *)		
ASS	Pre	34.65 ± 3.22	36.37 ± 1.94		
	Post	37.26 ± 2.40	37.60 ± 2.31		
	Post–Pre	2.61 ± 1.35	1.22 ± 0.72	−2.027	0.043 *
	z (*p*)	−2.521 (0.012 *)	−2.366 (0.018 *)		
Cadence	Pre	0.74 ± 0.10	0.70 ± 0.07		
	Post	0.81 ± 0.07	0.73 ± 0.08		
	Post–Pre	0.07 ± 0.04	0.02 ± 0.03	−2.296	0.022 *
	z (*p*)	−2.546 (0.011 *)	−1.706 (0.088)		

^a^ Mean ± SD, ASL: affected step length, SL: step length, ASS: affected single support, and * *p* < 0.05.

## Data Availability

The original contributions presented in this study are included in the article. Further inquiries can be directed to the corresponding author.

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
