# Peer review of "Gait Training with Virtual Reality-Based Real-Time Feedback for Chronic Post-Stroke Patients: A Pilot Study"

_healthcare, 2025, doi:10.3390/healthcare13020203_

Round 1
Reviewer 1 Report
Comments and Suggestions for Authors
Dear Authors,
your study explores a highly relevant topic, gait rehabilitation in post-stroke patients using virtual reality (VR)-based real-time feedback. This is a promising and innovative area, as VR technologies have the potential to enhance engagement and neuroplasticity in rehabilitation. Additionally, the study's focus on balance, gait ability, and motor function addresses critical outcomes for post-stroke recovery. The use of a control group is a strength, and the concept of integrating VR with treadmill training could provide an engaging alternative to traditional methods. However, significant methodological and reporting issues undermine the validity and impact of this work. The following points need to be addressed comprehensively:
Abstract
-
Background: the current section reads more like an Objective and does not provide sufficient context. Include a brief introduction to the use of VR in stroke rehabilitation and why it is important.
-
Objective: rewrite to clearly state the research aim (e.g., "to evaluate the effects of VR-based real-time feedback on motor function, balance, and spatiotemporal gait parameters in chronic stroke survivors").
-
Methods: the description is unclear and grammatically flawed. Include details about participants and study protocol and mention the statistical tests used.
-
Results and Conclusions: avoid overly strong claims such as "demonstrates." Replace with "suggests." Correct typographical errors like "Fugl-myear" to "Fugl-meyer."
Introduction
-
The introduction should clearly define the research gap. While virtual reality is a growing field in stroke rehabilitation, the rationale for combining VR with real-time feedback and treadmill training is not adequately discussed. Highlight why this specific combination is novel and necessary.
Methods
-
Terminology: replace "stroke patients" with "post-stroke" or "stroke survivors"
-
Time: Include a sentence about when the study was conducted.
-
Inclusion/Exclusion Criteria:
-
Rewrite as a clear, bulleted list. The criteria are currently confusing and incomplete.
-
Include critical details such as age range, stroke type (ischemic or hemorrhagic), stroke severity (NIHSS), spasticity, treatments received, and setting (patients residing at home/community setting vs. patients still hospitalized).
-
Address why factors such as ankle contractures are mentioned but not spasticity or other joint restrictions like at the hip or knee.
-
Procedure:
-
The procedure section lacks crucial details about how participants were recruited, assessed, and guided through the interventions.
-
Clarify the "standard rehabilitation program" and provide a detailed description of the control group's activities for fair comparison.
-
Describe the timing and qualifications of the personnel delivering the interventions and assessments.
-
"Procedure" and "Research Method" sections should be merged.
-
Motor function: there are no references. Please add them.
-
Sample Size and Confounders:
-
There is no mention of sample size calculation, stratification, or adjustment for confounders, which is a significant limitation.
-
The titles of the subsections in Section 2 should be revised for clarity. Suggested alternatives include: “Study Design,” “Population,” “Study Protocol,” “Outcome Measures,” and “Statistical Analysis.”
Results
-
Critical data such as stroke etiology (ischemic/hemorrhagic), spasticity, stroke severity (e.g., NIHSS), and time since stroke are missing.
-
The lack of consideration of confounders severely undermines the reliability of the results.
-
The small sample size (n=15) precludes meaningful statistical or clinical conclusions, especially given the heterogeneity of the participants.
Discussion
-
The discussion is overly optimistic and does not sufficiently address the study's limitations, including the small sample size and heterogeneity.
-
The findings are overstated, with terms like "demonstrated" needing to be replaced with "suggested" or "indicated."
-
There is minimal integration with broader literature on VR in stroke rehabilitation to contextualize the results.
The manuscript requires substantial revisions to the quality of the English language. There are numerous grammatical errors, awkward phrasings, and typographical mistakes that hinder clarity and readability. For example, "Fugl-myear" should be corrected to "Fugl-meyer," and sentences such as "In group analyses both groups showed significant improvements in motor function balance, gait ability and gait" need rephrasing for coherence. Professional English editing is strongly recommended to ensure the manuscript meets publication standards.
Author Response
Dear Editor
Thank you very much for reviewing our research.
We have thoroughly reviewed the content suggested by the Editor and revised it as follows.
Comments 1:
Abstract
Background: The current section reads more like an Objective and does not provide sufficient context. Include a brief introduction to the use of VR in stroke rehabilitation and why it is important.
Response 1:
The abstract has been organized by dividing sentences into contexts and adding background content to explain VR.
Comments 2:
Objective: Rewrite to clearly state the research aim (e.g., "to evaluate the effects of VR-based real-time feedback on motor function, balance, and spatiotemporal gait parameters in chronic stroke survivors").
Response 2:
We have written it clearly to state the purpose of this study and revised it to add content evaluating the effects of VR-based real-time feedback.
Comments 3:
Methods: The description is unclear and grammatically flawed. Include details about participants and study protocol and mention the statistical tests used.
Response 3:
Reviewed and corrected for grammatical errors and revised by mentioning statistical tests.
Comments 4:
Results and Conclusions: Avoid overly strong claims such as "demonstrates." Replace with "suggests." Correct typographical errors like "Fugl-myear" to "Fugl-meyer."
Response 4:
Claims such as "demonstrates." have been revised and replaced with "suggests." Also, "Fugl-meyer." has been revised.
Comments 5:
Introduction
The introduction should clearly define the research gap. While virtual reality is a growing field in stroke rehabilitation, the rationale for combining VR with real-time feedback and treadmill training is not adequately discussed. Highlight why this specific combination is novel and necessary.
Response 5:
The content has been revised to add the alternative content of the effect of treadmill training with VR real-time feedback.
Comments 6:
Methods
Terminology: replace "stroke patients" with "post-stroke" or "stroke survivors"
Response 6:
Since the flow of the sentence could change if all of it were changed to post-stroke, I used post-stroke only in the subject part, and since the title of the reference is stroke patients, there were limitations in editing.
Comments 7:
Time: Include a sentence about when the study was conducted.
Response 7:
Include a sentence about the study schedule.
Comments 8:
Inclusion/Exclusion Criteria:
Rewrite as a clear, bulleted list. The criteria are currently confusing and incomplete.
Include critical details such as age range, stroke type (ischemic or hemorrhagic), stroke severity (NIHSS), spasticity, treatments received, and setting (patients residing at home/community setting vs. patients still hospitalized).
Response 8:
This study included both ischemic and hemorrhagic strokes and included information on general characteristics. Since the study targeted all hospitalized patients, information on treatment and environment was not included.
Comments 9:
Address why factors such as ankle contractures are mentioned but not spasticity or other joint restrictions like at the hip or knee.
Response 9:
In the case of stroke patients, it is thought that unnatural changes in gait patterns occur due to ankle joint contractures, so ankle joint contractures were mentioned.
Comments 10:
Procedure:
The procedure section lacks crucial details about how participants were recruited, assessed, and guided through the interventions.
Response 10:
This study involved 15 post-stroke, regardless of gender, who were hospitalized at B Hospital located in Seongnam city, Gyeonggi Province. The study was conducted by recruiting participants who agreed to join after being introduced and informed about the study through a posted recruitment announcement.
Comments 11:
Clarify the "standard rehabilitation program" and provide a detailed description of the control group's activities for fair comparison.
Response 11:
Added description of the standard rehabilitation program.
Comments 12:
Describe the timing and qualifications of the personnel delivering the interventions and assessments.
Response 12:
Added personnel information for this study.
Comments 13:
"Procedure" and "Research Method" sections should be merged.
Response 13:
The procedure and intervention methods of this study were classified for readability because they were conducted in accordance with the latest healthcare journal format.
Comments 14:
Motor function: there are no references. Please add them.
Response 14:
Added references.
Comments 15:
There is no mention of sample size calculation, stratification, or adjustment for confounders, which is a significant limitation. Response 15:
The sample size of this study was calculated using the G-POWER program based on the results of previous studies.
Comments 16:
Critical data such as stroke etiology (ischemic/hemorrhagic), spasticity, stroke severity (e.g., NIHSS), and time since stroke are missing.
Response 16:
This study added ischemic and hemorrhagic etiology, and the severity and course of stroke were not investigated, so the limitations were added.
Comments 17:
The small sample size (n=15) precludes meaningful statistical or clinical conclusions, especially given the heterogeneity of the participants.
Response 17:
The heterogeneity of the participants due to the sample size was added to the limitations.
Comments 18:
The discussion is overly optimistic and does not sufficiently address the study's limitations, including the small sample size and heterogeneity. The findings are overstated, with terms like "demonstrated" needing to be replaced with "suggested" or "indicated."
There is minimal integration with broader literature on VR in stroke rehabilitation to contextualize the results.
Response 18:
In the discussion section of the results of this study, we added limitations and integrated the extensive literature on VR to minimize it. We also revised the term "demonstrated" to "suggested."
Reviewer 2 Report
Comments and Suggestions for Authors
Dear Authors,
The reviewed paper investigates a very important part of healthcare, the post-stroke rehabilitation.
Specifically, the authors examine the effectiveness of gait training with VR real-time feedback on motor function, balance, and spatiotemporal gait issues. This is why the article claims to be a valuable piece of research and an interesting read.
Please find below recommendations that might help improving the manuscript.
Introduction
Although the introduction describes general principles of VR, it lacks of explanation of VR implementation approaches. Recent studies show that VR can be used in clinical settings or as an autonomous home-based rehabilitative therapy for post-stroke patients, including telerehabilitation option. Telerehabilitation services remain increased post-COVID-19. The telerehabilitation treatment using VR improves stroke recovery because of the accessibility of the objective and measurable information. Moreover, selecting individualized VR exergames and VR-environment at the stage of planning of rehabilitation treatment allows adjusting regimen and improving balance training outcomes. The virtual therapist can guide patients, providing advices and encouragements, summarizing the results in the end of treatment session. Also, the VR system provides a variety of exergames with adjustable levels of difficulty to personalize the rehabilitation process. To strengthen the paper, it might be beneficial to explain shortly key approaches on using VR in motor function and balance problems recovery using traditional and telemedicine methods. This might help improving the justification of the study.
Please find below some reading that might help:
https://doi.org/10.1134/S2079057022030109
Methods
Readers would appreciate it if the authors mentioned whether the patients participating in the study were urban residents or people living in rural areas. Also, please explain who and how trained patients on the use of VR and related issues.
Results
Table 1:
Readers will appreciate if authors specify the stroke type among experimental and control groups.
Discussion
Line 337: … in the future, I believe …
Maybe to use “we” instead of “I” as there are three authors of the paper.
Conclusions
The paper might benefit if authors strengthen the conclusions by highlighting the most impactful outcomes of the research demonstrated the effects of a real-time feedback-based gait VR program on the spatial-temporal parameters of motor function, balance, and gait. Also, providing recommendations for prospective research, and for implementation of the findings in healthcare will be appreciated by healthcare professionals, and stakeholders.
In general, the paper is an interesting read that certainly brings a value to the field of post-stroke rehabilitation.
Regards,
Reviewer
Author Response
Dear Editor, Thank you for reviewing our study.
We will inform you of any corrections after reviewing the content.
Comments 1:
Introduction
Although the introduction describes general principles of VR, it lacks explanation of VR implementation approaches. Recent studies show that VR can be used in clinical settings or as an autonomous home-based rehabilitative therapy for post-stroke patients, including telerehabilitation option. Telerehabilitation services remain increased post-COVID-19. The telerehabilitation treatment using VR improves stroke recovery because of the accessibility of the objective and measurable information. Moreover, selecting individualized VR exergames and VR-environment at the stage of planning of rehabilitation treatment allows adjusting regimen and improving balance training outcomes. The virtual therapist can guide patients, providing advice and encouragements, summarizing the results in the end of treatment session. Also, the VR system provides a variety of exergames with adjustable levels of difficulty to personalize the rehabilitation process. To strengthen the paper, it might be beneficial to explain shortly key approaches on using VR in motor function and balance problems recovery using traditional and telemedicine methods. This might help improving the justification of the study.
Please find some reading below that might help:
https ://doi.org/10.1134/S2079057022030109
Response 1:
We reviewed helpful materials in this paper and added content on telerehabilitation, and tried to improve the validity of the study by reviewing existing content.
Comments 2:
Methods
Readers would appreciate it if the authors mentioned whether the patients participating in the study were urban residents or people living in rural areas. Also, please explain who and how trained patients on the use of VR and related issues.
Response 2:
VR-related therapist experience and The subjects were clearly stated. The patients participating in the study were urban residents. Comments 3:
Results
Table 1:
Readers will appreciate if authors specify the stroke type among experimental and control groups.
Response 3:
The table has been revised and reviewed.
Comments 4:
Discussion
Line 337: … in the future, I believe…
Maybe to use “we” instead of “I” as there are three authors of the paper.
Response 4:
Considering the number of authors on the paper, the expression was changed to "we".
Comments 5:
Conclusions
The paper might benefit if the authors strengthen the conclusions by highlighting the most impactful outcomes of the research demonstrated the effects of a real-time feedback-based gait VR program on the spatial-temporal parameters of motor function, balance, and gait. Also, providing recommendations for prospective research, and for implementation of the findings in healthcare will be appreciated by healthcare professionals, and stakeholders.
Response 5:
We have supplemented the conclusions and added information on the expected future effects of VR results.
In general, the paper is an interesting read that certainly brings a value to the field of post-stroke rehabilitation.
Dear Editor, Thank you for your careful review of our paper. I appreciate that you will continue to improve the quality of our research along with the development of our society.
Kim Sun Min, Professor, Department of Physical Therapy, Gimcheon University
Round 2
Reviewer 1 Report
Comments and Suggestions for Authors
Thank you for the revisions provided. However, several significant issues remain unresolved:
-
Abstract:
- Rephrase “this study was to” to indicate that the aim of the study is being discussed, not the study itself.
- Replace “randomized” with “randomly.”
- Revise the Results section to avoid repetition about gait: “In group analyses, both groups showed significant improvements in motor function, balance, gait ability, and gait.”
-
Methods (2.2 Subjects):
- Add "patients" after "post-stroke" to specify "post-stroke patients."
- Clarify whether the patients were inpatients or outpatients. Given the duration of illness (>6 months), it is more likely they were outpatients, but this must be explicitly stated based on what was actually done.
-
Inclusion Criteria:
- The 4th and 5th inclusion criteria are actually exclusion criteria, as they are expressed as negative clauses.
-
Response to Question 8:
- If the included patients were recruited at least 6 months post-stroke, this could include patients many years post-stroke. How can all these patients be hospitalized? If this is the case, it must be clearly described in the text, as it appears unlikely otherwise.
-
Response to Question 9:
- This response is weak and fails to address critical clinical issues like post-stroke spasticity (PSS), a common and debilitating condition that strongly impacts and affects gait and other motor functions in chronic stroke patients.
-
Comment 14:
- There is still no bibliographic reference for section 2.5.1 on motor function.
-
Sample Size Calculation:
- Move the sentence about sample size calculation to paragraph 2.6.
- The other criticisms in Question 15 remain unaddressed.
-
Comment 16:
- There is no answer regarding spasticity or the mean time since stroke.
-
Table 1:
- Correct the etiologies of stroke from “infarction” and “hemorrhagic” to “ischemic” and “hemorrhagic.”
- Add the missing data to the table, as only the title has been provided.
The quality of English in the manuscript requires improvement. There are issues with repetitive phrasing, awkward sentence construction, and unclear expressions, particularly in key sections such as the Abstract and Results.
Author Response
Command 1:
Abstract:
Rephrase “this study was to” to indicate that the aim of the study is being discussed, not the study itself.
Response 1:
This study aimed to investigate the effectiveness of gait training with virtual reality-based real-time feedback on motor function, balance, and spatiotemporal gait parameters in poststroke patients. The correction has been completed in the form of a sentence.
Command 2:
Replace “randomized” with “randomly.”
Response 2:
The sentence was modified to randomly.
Command 3:
Revise the Results section to avoid repetition about gait: “In group analyzes, both groups showed significant improvements in motor function, balance, gait ability, and gait.”
Response 3:
“In group analyses, both groups showed significant improvements in motor function, balance, gait ability, and gait.” The content was deleted because gait was duplicated in the sentence.
Command 4:
Methods (2.2 Subjects):
Add "patients" after "post-stroke" to specify "post-stroke patients."
Clarify whether the patients were inpatients or outpatients. Given the duration of illness (>6 months), it is more likely that they were outpatients, but this must be explicitly stated based on what was actually done.
Inclusion Criteria:
Response 4:
All sentences have been revised to "post-stroke patients."
Added information on the classification of patients by duration of hospitalization and outpatient care.
This study was conducted with participants who agreed to participate after being introduced to and informed about the research through a recruitment announcement. The participants were outpatients who had experienced a stroke for more than six months.
Command 5:
Inclusion Criteria:
The 4th and 5th inclusion criteria are actually exclusion criteria, as they are expressed as negative clauses.
Response to Question 8:
If the included patients were recruited at least 6 months post-stroke, this could include patients many years post-stroke. How can all these patients be hospitalized? If this is the case, it must be clearly described in the text, as it appears unlikely otherwise.
Response to Question 9:
Response 5:
Response to Question 8: Since questions 4 and 5 mean that those without clinical symptoms should be included, I would like to say that there is no problem with the inclusion criteria. Thank you.
Response to Question 9: Based on the editor's opinion, I have specified that they are outpatients. Thank you for your good opinion. The hospital where this study was conducted was a general hospital, and the study was conducted on outpatient stroke patients who were discharged after hospitalization. I have added the information that it was conducted on outpatients in the text. Thank you.
Command 6:
This response is weak and fails to address critical clinical issues like post-stroke spasticity (PSS), a common and debilitating condition that strongly impacts and affects gait and other motor functions in chronic stroke patients.
Response 6:
We will accept all of the editor's comments and use a standardized assessment tool that can assess stroke severity (NIHSS) or the degree of spasticity before the experiment to pre-measure the degree of spasticity in future studies. We will add the content that we were unable to perform this spasticity assessment to the limitations section.
" Fourth, the sample size was small, and the severity of the patients' strokes and the degree of spasticity were not carefully assessed in the analysis. Future studies should include larger sample size and an assessment of spasticity to better generalize the effectiveness of virtual reality interventions for stroke patients."
Command 7:
There is still no bibliographic reference for section 2.5.1 on motor function.
Response 7:
References to 2.5.1 have been added. thank you
Command 8:
Sample Size Calculation:
Move the sentence about sample size calculation to paragraph 2.6.
Response 8:
The edit has been completed.
Command 9:
The other criticisms in Question 15 remain unaddressed.
Response 9:
Thank you for your good opinion. I calculated the number of subjects using the G-power program, and conducted the experiment with a total of 15 subjects. Thank you.
Command 10 :
There is no answer regarding spasticity or the mean time since stroke.
Table 1:
Correct the etiologies of stroke from “infarction” and “hemorrhagic” to “ischemic” and “hemorrhagic.”
Add the missing data to the table, as only the title has been provided.
Response 10 :
Added the missing data to the table and added data on the mean time since stroke. Thank you.
Dear Editor-in-Chief,
Thank you very much for your careful review of our original text from beginning to end.
We, the authors, feel that the quality of our original text has improved thanks to your feedback. We will keep this in mind for a long time.
Have a Merry Christmas and a Happy New Year.
Professor Sun Min Kim, Department of Physical Therapy, Gimcheon University
Reviewer 2 Report
Comments and Suggestions for Authors
Dear Authors,
The paper has been improved.
Regards,
Reviewer
Author Response
Dear Editor-in-Chief
We sincerely thank you for your careful review of our paper.
We pray for the endless development of the Healthcare Journal.
Have a happy day today.
- Professor Kim Sun Min, Department of Physical Therapy, Gimcheon University -
Round 3
Reviewer 1 Report
Comments and Suggestions for Authors
Thank you for the revisions you made.